# Immune Diseases Associated with Aging: Molecular Mechanisms and Treatment Strategies

**DOI:** 10.3390/ijms242115584

**Published:** 2023-10-25

**Authors:** Mi Eun Kim, Jun Sik Lee

**Affiliations:** Department of Biological Science, Immunology Research Lab & BK21-Four Educational Research Group for Age-Associated Disorder Control Technology, Chosun University, Gwangju 61452, Republic of Korea; kimme0303@chosun.ac.kr

**Keywords:** aging, immunosenescence, immune diseases, molecular therapies

## Abstract

Aging is associated with a decline in immune function, thereby causing an increased susceptibility to various diseases. Herein, we review immune diseases associated with aging, focusing on tumors, atherosclerosis, and immunodeficiency disorders. The molecular mechanisms underlying these conditions are discussed, highlighting telomere shortening, tissue inflammation, and altered signaling pathways, e.g., the mammalian target of the rapamycin (mTOR) pathway, as key contributors to immune dysfunction. The role of the senescence-associated secretory phenotype in driving chronic tissue inflammation and disruption has been examined. Our review underscores the significance of targeting tissue inflammation and immunomodulation for treating immune disorders. In addition, anti-inflammatory medications, including corticosteroids and nonsteroidal anti-inflammatory drugs, and novel approaches, e.g., probiotics and polyphenols, are discussed. Immunotherapy, particularly immune checkpoint inhibitor therapy and adoptive T-cell therapy, has been explored for its potential to enhance immune responses in older populations. A comprehensive analysis of immune disorders associated with aging and underlying molecular mechanisms provides insights into potential treatment strategies to alleviate the burden of these conditions in the aging population. The interplay among immune dysfunction, chronic tissue inflammation, and innovative therapeutic approaches highlights the importance of elucidating these complex processes to develop effective interventions to improve the quality of life in older adults.

## 1. Introduction

Aging is a universal biological phenomenon that continues throughout life and directly affects the immune system. The immune system, a regulator of several critical biological processes, interacts with the neurological, circulatory, and other systems to protect against internal and external diseases [1]. The immune system and its reactions are significantly affected by aging, a complex process. Increased susceptibility to infectious diseases, a relatively inferior response to immunization, an increase in the prevalence of cancers and autoimmune disorders, and other chronic diseases characterized by pro-inflammatory states are symptoms of immune system deterioration with age [2]. Immunosenescence is a term used to describe age-related changes in cellular and serological immune responses that affect how the body produces specialized responses to self and foreign antigens. Immunosenescence is an aging process that results in a decline in the function, number, and efficiency of the immunological organs, immune cells, and cytokines [3,4]. In addition to the decreased activity of antigen-presenting cells (APCs), as we age, there is a decrease in the number and regulatory capacity of T cells, which impairs infection response and tumor control. Simultaneously, B cells decrease in number, leading to decreased antibody production, thereby hindering defense against infection. Telomere shortening contributes to B-cell senescence. Senescent immune cells release pro-inflammatory molecules, collectively known as the senescence-associated secretory phenotype (SASP), contributing to chronic inflammation associated with age-related diseases. The main causes of the SASP are related to DNA damage, inflammatory and cellular stress signaling, impaired cell cycle control, altered gene expression, and decreased mitochondrial function. These changes involve multiple mechanisms, such as telomere shortening, inflammation, and signaling pathways, and therefore require a comprehensive understanding of immune senescence. Immunosenescence increases the prevalence of age-related disorders, including cardiovascular diseases, neurological diseases, malignancies, and autoimmune diseases [5,6,7,8]. Immunosenescence is a multifactorial and dynamic complicated phenomenon characterized by a protracted adjusting and remodeling process in the immune system throughout life [1]. Herein, we review immune diseases associated with aging and treatment methods based on the underlying molecular mechanisms.

## 2. Immune Diseases Associated with Aging

### 2.1. Cancer

Age-related decrease in immune function, or immunosenescence, may render it relatively more difficult for aged people to produce robust immunological responses to antigens, including defenses against tumors [9]. This behavior fits with the theory of cancer immunoediting, according to which immune-evading and poorly immunogenic-altered cells can evade immune monitoring. Subsequent research has revealed a complex model of immunosuppression induction by tumors, which has many different aspects. The immune system is suppressed via various strategies employed by established tumors, such as the secretion of immune-suppressive cytokines, the expression of immune-suppressive chemicals, and the production of regulatory cells [10,11,12]. Additionally, the immune system undergoes several age-related changes known as immunosenescence, including reductions in B- and T-cell proliferation and function, changes to cellular subsets, functional limitations, and qualitative changes in APCs [13]. Immunosenescence impairs the innate immune response and interferes with several activities of APCs. These outcomes include diminished pro-inflammatory cytokine release, decreased expression of toll-like receptors, and decreased antigen processing capacity owing to age-related proteasome dysfunction. Age-related decrease in the ability of APCs to label tumor antigens is evident in the decline in immunologic function [14]. Reduced immune cell variety brought on by these alterations may result in weakened immunological responses to infections and cancerous cells. In addition, as the number of naive CD8+ T cells declines and that of antigen-exposed CD4+ and CD8+ T cells rises with age, the ability to neutralize freshly encountered antigens is diminished. The abundance of apoptotic molecules like CD95 and costimulatory molecules like CD28 and CD27 on T cells decreases with age [14]. The deterioration in T-cell immune function occurs owing to a reduction in the number of these molecules, which directly impacts the immunological activity of T cells. In the case of T cells, for example, different T-cell subsets can be adversely regulated by tumors, and the intra-tumoral increase in the number of regulatory T cells (Tregs) is linked to a poor prognosis in various cancer types. Chemokines released by tumors make it easier for particular Tregs to enter the tumor environment. In addition, tumors produce several pro-inflammatory cytokines that activate and grow myeloid-derived suppressor cells and Tregs. The importance of Tregs in promoting metastatic growth has been underlined by studies using breast cancer model organisms. However, the inflammatory environment characteristic of older adults may impact the likely cumulative effects of cancer and aging in causing the growth of particular subsets of Tregs, thereby encouraging tumorigenesis and metastasis. The increase in the number of Tregs is another factor that accelerates immunological aging [15]. The majority of these modifications affect the makeup of T lymphocytes. Thymic involution in older adults results from the loss of activity of naive T cells. Parallel to this, persistent antigenic stimulation causes naive T cells to transform gradually into virtual memory cells [15]. In addition, the tumor microenvironment can depress the immune system by preventing hematopoietic stem cells from differentiating into mature dendritic cells (DCs). The maturation and functionality of DCs are hampered by tumor-associated stimuli, whereas the growth of Tregs is encouraged. This cytokine-rich milieu may limit or eliminate the activation of natural killer (NK) cells and cytotoxic T lymphocytes, thereby enhancing immunological evasion. Apoptotic cells can activate macrophages and DCs, causing the release of anti-inflammatory mediators that further regulate immunosuppression and promote tumor growth [15,16,17]. For instance, tumor necrosis factor (TNF)-α and interleukin (IL)-1α can inhibit apoptosis, promoting the development of cancer cells. In addition, these cytokines may play a role in immunosuppressive processes [18,19]. In both aging and cancer, IL-6 upregulation, which occurs through various pathways, can contribute to immunosuppression [20]. These results raise the possibility of reversing immunosuppression in elderly cancer patients, whereas the relative significance of age-related immunosuppression in cancer genesis and progression is yet unknown. Interest in utilizing anti-inflammatory medications to prevent cancer has stemmed from research on aging, inflammation, and cancer. Preclinical research and clinical evidence point to the viability of using anti-inflammatory methods to protect against cancer in older adults [14]. These characteristics are essential components of the complex process of immunosenescence, emphasizing the subtle adjustments that the immune system undergoes as people age.

### 2.2. Atherosclerosis

Aging is a major risk factor for atherosclerosis, a condition characterized by the buildup of plaques within arterial walls. Immune cells, particularly macrophages, contribute to plaque formation and progression [21]. Senescent immune cells and altered signaling pathways in aging vessels promote chronic inflammation and plaque instability [22]. Atherosclerosis is a disorder associated with aging that could be aggravated by inflammation [23]. Immune cells eliminate senescent cells to prevent inflammatory responses. Contrastingly, senescent immune cells are relatively less effective, resulting in a buildup and persistence of SASP release, which hastens aging and age-related diseases. SASP molecules, secreted by senescent cells, comprise various biological compounds, including inflammatory cytokines, growth factors, and chemokines such as C-C chemokine motif ligand (CCL)-2 and CCL-20. These molecules collectively constitute an “inflammatory secretome”. Excessive SASP release from senescent cells contributes to age-related diseases by promoting chronic inflammation and damaging neighboring cells. Senescent immune cells are present in the arterial wall and can be identified by an increase in the abundance of pro-inflammatory mediators, which aggravate atherosclerotic plaque development and are released by macrophages, DCs, and foam cells [24,25,26]. Senescent immune cells may aid in decreasing necrosis and plaque development. Decreased lymphocyte receptor diversity and weakened immune cell responses due to aging may contribute to the induction of atherosclerosis [27,28,29]. Senescent immune cells have several characteristics in common, including DNA damage, reduced gene and mitochondrial function, cell cycle arrest, apoptosis resistance, and the synthesis of SASP as inflammatory mediators [30]. The contribution of the link between SASP release and the senescence of DCs to atherosclerosis remains elusive. However, the ability of DCs to bind with T cells is known to decline with age. Reduced chemotaxis, phagocytosis, and antigen presentation to the effector T cells are a few of these alterations, along with a dearth of peripheral DCs. In addition, the development of cardiovascular diseases is linked to a chronic inflammatory condition caused by the co-occurrence of aging and the atherogenic response [24]. Senescent T cells express interferon (IFN)-γ, C-C chemokine receptor (CCR)-7, and IL-18, which in turn boost the number of CD4+/CD8+ cells (effector memory cells re-expressing CD45RA, termed TEMRA). These cells activate the p38 mitogen-activated protein kinase (MAPK) signaling pathway. The atherosclerotic plaque includes numerous senescent TEMRA cells [31]. The overactivation of the p38-MAPK signaling pathway causes T-cell mediated IFN-γ production and T-cell receptor (TCR) signaling, exacerbating the development of atherosclerotic plaque [32,33]. Additionally, it has been shown that senescent CD4+/CD28+ T cells are linked to IFN-γ secretion, and individuals with unstable angina have increased levels of this subgroup. Senescent CD4+/CD28+ T cells release large amounts of C-X-C chemokine receptor (CXCR) 1, CCR5, and CCR7, which encourages inflammation at the location of the atherosclerotic plaque [34].

As reported in a recent paper, distinct subsets of Tregs have a significant role in influencing the development and progression of atherosclerosis. Tregs are instrumental in maintaining immune tolerance and curbing excessive immune responses. These subsets of Tregs exert their effects by mitigating the responses of effector T cells, thus impacting atherosclerosis [35,36]. The deficiency in Treg function or development can exacerbate the formation of atherosclerotic lesions [35]. Additionally, the discussion delves into another category of Tregs known as inducible Tregs (iTregs). This group includes type 1 Tregs (Tr1) and Tregs that produce transforming growth factor-beta (TGF-β), designated as Th3 [37]. Unlike natural Tregs, iTregs develop from naive or effector T cells in peripheral tissues such as the intestine. Despite being CD4- and CD25-positive, iTregs do not require Foxp3 for their functionality. Tr1 Tregs secrete IL-10 and Th3 Tregs produce TGF-β [38]. These iTregs play a vital role in maintaining the immune balance by secreting immunosuppressive cytokines, thereby inhibiting atherosclerotic lesion formation [39]. Both natural Tregs and iTregs can prevent autoimmunity and excessive immune responses by suppressing the proliferation and differentiation of naive and effector T cells [40,41,42]. Irrespective of their antigen specificity, Tregs can exert suppressive functions post-activation via antigen presentation. Moreover, both inducing polyclonally activated Tregs and antigen-specific Tregs could be potential therapeutic avenues for addressing atherosclerosis [43]. A clinical study reported that patients with coronary artery disease display decreased levels of circulating natural Tregs and a reduced Treg-to-effector T-cell ratio than that in controls [44,45]. Earlier research indicated a connection between lower levels of Foxp3-positive cells and a higher risk of coronary events. Considering previous investigations, a recent report suggests that enhancing the Treg-to-effector T-cell ratio through the suppression of effector T-cell responses and the augmentation of Treg numbers or activities could hold promise as a therapeutic strategy for atherosclerotic cardiovascular disease [46]. Moreover, it has been shown that IL-2 plays a crucial role in T-cell proliferation and differentiation, particularly in maintaining the abundance and functionality of Tregs. Administering a recombinant mouse IL-2/anti-IL-2 monoclonal antibody complex leads to selective expansion of CD4+ CD25+ Foxp3+ Tregs [47]. This IL-2 complex therapy demonstrated the potential to attenuate atherosclerosis and inhibit aneurysm development by fostering the expansion of specific populations of Tregs.

### 2.3. Immunodeficiency Disorders

Older adults frequently have impaired immune systems, making them relatively more vulnerable to illnesses. Immunodeficiency is primarily caused by thymic involution and a reduction in T-cell diversity [48]. The senior population is more prone to be severely affected by infections like influenza, pneumonia, and herpes zoster. The most prevalent primary immunodeficiency is common variable immunodeficiency (CID). Although the molecular pathophysiology of this condition is not entirely understood, a few patients with dominant inheritance have genetic predispositions, and the immunodeficiency is diagnosed based on reduced levels of IgG, in addition to those of IgA, IgM, or both. A functional impairment of T cells is present in nearly one-third of patients [49], as immunological failure in CID affects T and B cells and DCs. The primary pathophysiological mechanism underlying the condition is the inability of B cells to develop into plasma cells, which leads to decreased Ig production. Although the total number of B cells may be normal, the cells themselves are not. The adaptive immune system is weakened owing to the decrease in Ig production, and patients are relatively more likely to have recurring infections, with up to 30% of patients with CID developing autoimmune processes, of which 2.3% may have autoimmune disease as their sole clinical finding in CID diagnosis [50]. Common autoimmune conditions in CID patients include autoimmune hemolytic anemia, immune thrombocytopenic purpura, Evan’s syndrome, which is a combination of autoimmune hemolytic anemia and immune thrombocytopenic purpura, and seronegative rheumatoid arthritis (RA). However, a few autoimmune diseases like type I diabetes mellitus and seropositive RA are not aggravated in CID [50]. Autoimmunity can be the initial presentation of CID without typical recurrent infections. Patients with fewer switched memory B cells or granulomatous infiltrations are relatively more likely to develop autoimmune issues [51]. Consequently, the body becomes relatively more vulnerable to infection and disease, owing to modifications in the immune system caused by aging. These modifications include decreased T- and B-cell diversity and widespread immunodeficiency.

## 3. Molecular Mechanisms Underlying Immune Aging

### 3.1. Telomere Shortening

Cellular senescence is caused by the shortening of telomeres during cell division. Both immune and nonimmune cells are impacted by this process, which adds to the general deterioration of tissue homeostasis and immunological function [52]. At the ends of linear chromosomes, telomeres are protein–DNA complexes of short, tandem G-rich hexanucleotide repeats and related proteins [53]. With each cell division, their length decreases, which correlates inversely with age. Reactive oxygen species (ROS), inflammatory responses, genetic and epigenetic variables, and sex-related hormones can all alter the lengths of telomeres [54]. A cell cycle arrest or senescence is caused by telomeres shorter than a critical minimum length. Immune system competence solely depends on cell renewal and the clonal proliferation of B- and T-cell populations and therefore is extremely vulnerable to telomere shortening. The ability of immune system cells to increase telomerase, an enzyme that lengthens telomeres, and to prevent telomere attrition during cell proliferation makes them distinct from other types of somatic cells [55]. Humans have an overly wide range of telomere length. The population of CD4+ and CD8+ T lymphocytes, B lymphocytes, granulocytes, monocytes, and NK cells all showed lineage-specific telomere shortening with various telomere attrition kinetics [56].

### 3.2. Immunosenescence

Immunosenescence is characterized by a unique immune system remodeling caused by antigen exposure and oxidative stress. Due to a steady loss of naive T and B cells and a reduction in the absolute numbers of T and B lymphocytes, adaptive immunity deteriorates in the aging immune system. Despite certain age-dependent changes being visible, the innate compartment of the immune system is rather well conserved. As most of their immunological markers retain their appropriate levels, nonagenarians and centenarians provide a successful example of immune system aging [57]. Moreover, in vitro and ex vivo, neutrophils detrimentally impact telomeres in a manner contingent upon ROS. Neutrophil depletion mitigates senescence and ameliorates telomeric perturbations in a murine model of acute hepatic injury. The recruitment of neutrophils to the aging liver is orchestrated by senescent cells, thereby proposing a plausible mechanism for the propagation of senescence to adjacent cellular entities [58].

### 3.3. Inflammation and SASP

Senescent immune cells secrete pro-inflammatory cytokines, chemokines, and matrix metalloproteinases, collectively known as the SASP. The chronic inflammation induced by the SASP disrupts tissue microenvironments and contributes to the pathogenesis of various age-related diseases [59]. The SASP describes the production of several cytokines, chemokines, growth factors, proteases, and lipids by senescent cells. Depending on the senescence trigger, the composition of this unique secretome changes. The SASP has certain positive effects, including enabling the immune system’s recruitment to premalignant lesions and encouraging the repair of injured tissues [60,61,62]. The secretion of numerous pro-inflammatory substances, including IL-6, IL-8, membrane cofactor proteins (MCP), and macrophage inflammatory proteins (MIP) [63], can however have detrimental effects by enhancing proliferation, angiogenesis, and inflammation in both autocrine and paracrine ways. The SASP has been implicated in a DNA-damage signaling mechanism [64]. In hematopoietic cells, it led to the accumulation of spontaneous DNA damage in immune cells and increased expression of senescence and SASP markers in several cell types, e.g., B, NK, and CD4+ and CD8+ T cells and monocytes/macrophages, similar to what was seen in 2-year-old naturally aged mice [65,66]. These senescent immune-cell-aged animals displayed signs of aging, including decreased lifespan and multi-organ tissue degradation. The level of senescence in nonimmune organs such as the liver, kidney, intervertebral discs, and muscle increased in the animals [52]. Likewise, a shorter-than-normal lifespan and the induction of senescence in several tissues were caused by the adoptive transfer of splenocytes from old wild-type mice into young mice [65]. These findings imply that DNA damage can cause immune cells to become more senescent as they age, which can subsequently trigger secondary senescence by secreting SASP factors. However, the precise immune cell subsets that promote aging and systemic senescence in lymphoid and nonlymphoid organs must be identified. The adoptive transfer of young immune cells into a mouse model of accelerated senescence and aging, however, led to a decrease in the number of senescent cells in different tissues, indicating that young immune cells are capable of removing senescent cells that develop as a result of aging and disease [67]. In the case of B cells, it has been demonstrated that adipose tissue B cells display a senescence signature, including increased SASP, than that by B cells in the blood. In addition, memory B cells preferentially express SASP markers in the periphery [68], with age-associated B cells (ABCs), a subpopulation of B cells that increase with aging in mice, on the rise. These cells have been identified phenotypically as B220-CD19+ and negative for the mature B-cell markers CD21 and CD23. In addition, T-box transcription factor (T-bet) and CD11c expression have been observed in ABCs. These B cells develop a pro-inflammatory phenotype as a result of T-bet. Increased TNF-α and IFN-γ production from ABCs promotes CD4+ T-cell development into Th17 cells. Resident ABCs release TNF-α, which indicates their ability to inhibit B-cell growth in the bone marrow [65,69]. These results suggest that the senescence-like phenotype in B cells, and likely other immune cell types, depends on their age-associated function.

### 3.4. Altered Signaling Pathways

Age-related changes in signaling pathways, such as the mammalian target of rapamycin (mTOR) pathway, contribute to immune cell dysfunction. Dysregulated nutrient sensing and metabolism impact T-cell activation, differentiation, and immune responses [70]. The mTOR kinase belongs to the PI3K-related kinase (PIKK) family of proteins, which acts as a fundamental regulator of cellular growth and metabolism in response to nutritional and hormone stimuli [71]. In addition, mTOR is a key regulator of aging in organisms ranging from yeast to mammals. Pharmacological blocking of the mTOR complex (mTORC) 1 pathway using rapamycin or S6K1 mutant mice, both of which enhanced lifespan, demonstrated the importance of this mechanism in regulating lifespan [72]. The inhibitory effect of mTORC1 on autophagy is well known, and it has been suggested that the decline in autophagy with age is driven by increased mTORC1 activity. However, senescent CD8+ T cells demonstrated low autophagy and no mTORC1 activity. Rapamycin did not affect autophagy in senescent CD8+ T cells [71]. Aging is associated with a low-grade chronic inflammatory state, and mTOR hyperactivation is frequently related to inflammation. Contrastingly, the enhanced inflammation reported in senescent CD8+ T cells is caused by elevated p38 MAPK activity, as this kinase is implicated in producing pro-inflammatory cytokines [73]. Human studies are being conducted to investigate the effects of mTOR inhibition on aging, with a recent study showing that the mTOR inhibitor RAD001 increased antibody titers to influenza vaccination and lowered programmed death protein (PD) 1 expression on CD4+ and CD8+ T cells [74]. Notably, PD-1 is an inhibitory receptor that has received considerable attention in the pharmaceutical sector, with PD1/PD-1 ligand (PD-L1) drugs effectively treating melanoma, renal carcinoma, and non-small cell lung cancer [71,75]. mTORC1 signals are required to initiate the cell cycle and synchronize the initial metabolic changes following T-cell activation. Raptor-deficient T cells fail to upregulate the expression of Glut1 or other glycolytic enzymes during their activation, impairing lipid synthesis. mTORC1 governs the metabolism of effector T-cell development apart from its role in T-cell activation [76,77]. Once mTORC1 is knocked out, the expression of numerous mitochondrial metabolism-related genes, including fatty acid oxidation-related transcripts, increases. CD8 memory T cells lacking in Rictor, a key component of mTORC2, may improve overall metabolic capability, as seen by increased glycolysis and mitochondrial spare respiratory capacity (SRC) [76,78]. The development of variants capable of manipulating mTORC1/mTORC2 activity includes three generations of mTOR inhibitors that have been created due to the crucial regulatory activities of mTOR in cell growth and metabolic control [79]. Rapamycin and its derivatives are the first generation of mTOR inhibitors, which target mTOR and FKBP12 to modify the conformation of mTOR and thereby reduce the kinase activity of mTORC1. The second generation of mTOR inhibitors, including TORIN, are ATP competitive inhibitors that inhibit both mTORC1 and mTORC2 [76,79]. Third-generation mTOR inhibitors include the Rictor-mTOR interaction blocker JR-AB2-011 and the linked Rapalog-ATP competitor RapaLink-1, both of which demonstrate increased target selectivity or efficacy [76]. Manipulation of these pathways has been reported to endow CD8+ T cells with greater antitumor ability, consistent with the roles of mTORC1/mTORC2 in dictating CD8+ T-cell fate; rapamycin pretreatment of CAR-T cells during ex vivo expansion reduced mTORC1 activity while increasing CXCR4 expression, allowing these cells to infiltrate bone marrow and reduce resident AML cells [80]. Another study verified rapamycin’s therapeutic potential in CAR-T therapy by demonstrating that the combination of IL-2 and rapamycin caused human CART cells to develop memory stem cell characteristics. Given the impact of the mTORC2 signaling system, cholesterol-lowering treatment may decrease mTORC2 activity in CD8+ T cells, resulting in better tumor control [76,81]. It has been proposed that targeting mTOR signaling could improve the efficacy of PD-1-targeted treatment. PD-1-targeted treatments improve the T-cell response to chronic infection and malignancies by encouraging the differentiation of progenitor Tex (exhausted) cells into effector cell subsets. Moreover, PD-1 signals are required to regulate the essential balance of mTOR-dependent anabolic glycolysis and fatty acid oxidation programs to meet the bioenergetic needs of quiescent CD8+ memory T cells [82]. Consequently, by applying mTOR to regulate the activity and differentiation of T cells, it will be possible to control the age-related immune response.

Thus, a thorough understanding of immunosenescence, telomere shortening, the SASP, and alternate signaling pathways could be useful in understanding the molecular mechanisms controlling immunological aging (Table 1).

## 4. Treatment Strategies Based on Molecular Mechanisms

### 4.1. Targeting Inflammation Using Probiotics

Anti-inflammatory medicines, such as corticosteroids and nonsteroidal anti-inflammatory drugs, are routinely used to treat immunological illnesses in the elderly. Clinical trials of novel medicines targeting individual cytokines or signaling pathways in the inflammatory cascade show promise. In recent years, diverse therapeutic approaches have emerged that involve the utilization of probiotics for anti-inflammatory regulation and the modulation of immune cells. Probiotic strains have been found to have a variety of different impacts on the host and its immune system. The failure of the systemic immune regulatory networks, which sets off a chain of events that results in an inflammatory response, has been demonstrated in several in vitro and ex vivo models and animals, illuminating their crucial function in the regulation of inflammation [83]. Several bacterial strains can influence the intestinal mucosal barrier, the gut luminal environment, and the mucosal immune system. Probiotics may have an impact on a variety of innate and acquired immune system cells, including monocytes, macrophages, DCs, NK cells, and lymphocytes [84].

Studies have shown that the anti-inflammatory cytokine released in the stomach may be the cause of the immune regulation resulting from probiotic bacteria. In the context of aging and inflammation, a delicate balance between beneficial and potentially harmful microorganisms in the gut can be disrupted, often accompanied by a decrease in intestinal biodiversity. This disruption can lead to gut inflammation and a condition known as ‘intestinal dysbiosis’. Intestinal dysbiosis involves an imbalance in the complex interactions between the immune system, the gut neuroendocrine system, and the mucosal barrier. It can result in acute inflammatory reactions triggered by events such as gastrointestinal viruses or bacterial infections. More commonly, dysbiosis develops gradually due to underlying causes, leading to a chronic condition associated with various age-related disorders affecting the digestive system and other organs and systems [82,84]. However, the precise molecular interactions between the host and the probiotics are not well understood. The three genera Lactobacillus, Bacillus, and Bifidobacterium are the most often utilized probiotics in humans; however, the genus Saccharomyces is widely employed in commercial goods [85]. The generation of cytokines by immune cells may be modified by particular Lactobacillus strains, and Bifidobacterium induces the development of tolerance [84]. Each probiotic strain exhibits such diverse regulatory functions, which are related to their structure, the range of mediators they release, and the numerous pathways they simultaneously engage. Probiotics have been examined for their anti-inflammatory properties in vitro, ex vivo, and in animal tests to assess cytokine production and immune cell proliferation [83]. The masters of immune regulation and tolerance among the other immunological players are Tregs. Both Lactobacillus reuteri and Lactobacillus casei exert anti-inflammatory effects by increasing the levels of IL-10 and the activation of Tregs [83]. Overall, the pro-inflammatory cascade is markedly downregulated, and the proliferation of bystander T cells is inhibited. In vivo investigations on inflammatory illnesses, such as Crohn’s disease and atopic dermatitis, have employed these strains. A combination of probiotics (*Bifidobacterium bifidium*, *Lactobacillus acidophilus*, *L. casei*, *L. reuteri*, and *Streptococcus thermophiles*) that can reduce B- and T-cell responses with a net production of inhibitory cytokines has been shown to have this activity on Tregs [86]. Novel probiotics *L. rhamnosus* GG and *L. reuteri* DSM 17938 block the activation of T cells and NK cells and the release of IFN-γ from Staphylococcus aureus-cultured PBMCs, indicating that certain strains can either negatively or positively stimulate NK cells [87]. Most patients with inflammatory bowel disease (IBD) have a deep dysbiosis compared with that in healthy persons, with decreased diversity, a reduction in anti-inflammatory taxa, and an increase in Proteobacteria, Fusobacteria, and Pasteurellaceae. The increased gut permeability that is common in IBD patients makes it easier for various bacteria to get through the intestinal layer [88]. The interaction of the microbiota with epithelial cell receptors causes a chronic inflammation that affects the disease [83]. While probiotics capable of enhancing IBD have not yet been formulated, the possibility of their development has been suggested through several studies.

### 4.2. Targeting Inflammation Using Natural Compounds

In addition, natural compounds are currently being employed in the therapeutic intervention of an additional age-associated immune disorder. Polyphenols have a wide range of applications in the therapeutic management of immunological problems linked to aging because they possess anti-inflammatory and antioxidant properties. Polyphenols are a type of natural product found in foods such as fruits, green tea, coffee, and vegetables [89].

The polyphenol curcumin is taken out of the rhizome of the turmeric plant, Curcuma longa. Through interactions with a variety of molecular targets, curcumin exhibits antioxidant, anti-inflammatory, and anti-cancer activities in numerous disease models. As a result, it possesses a variety of pharmacological properties [90]. Curcumin’s ability to decrease nuclear factor kappa B (NF-κB), a crucial transcription factor that controls the expression of numerous genes involved in the immune and inflammatory responses, may contribute to its anti-inflammatory effects. Through the inhibition of NF-κB signaling, curcumin reduces the expression of pro-inflammatory cytokines and cyclooxygenase-2 (COX2). Curcumin exerts protective effects against atherosclerosis by influencing the polarization of macrophages, causing a shift from pro-inflammatory M1 macrophages to the anti-inflammatory M2 phenotype. This effect is achieved through its modulation of Toll-like receptor (TLR) 4, MAPK, and NF-κB pathways [91,92]. In addition, curcumin reduces airway inflammation in an asthma model induced by ovalbumin (OVA), leading to decreased levels of inflammatory cytokines (IL-4, IL-5, TNF-α) and fewer inflammatory cells (eosinophils, neutrophils, lymphocytes) [93]. This reduction is attributed to its actions on the MAPK and NF-κB pathways. Furthermore, curcumin mitigates inflammation and airway remodeling in a model of chronic obstructive pulmonary disease (COPD) induced by lipopolysaccharide (LPS) and cigarette smoke [94]. It achieves this by inhibiting NF-κB activation and COX-2 expression, resulting in decreased levels of TGF-β and IL-6 and reduced infiltration of inflammatory cells.

Quercetin, a polyphenolic compound belonging to the flavonoid class, is abundantly present in various fruits and vegetables and in tea [95]. In the context of research, quercetin has demonstrated anti-arthritic effects in rats with collagen-induced arthritis (CIA) by influencing the balance between Th17 cells and Tregs. Its effects encompass the inhibition of the nucleotide-binding domain 3 (NLRP3) inflammasome activation and the induction of an anti-inflammatory response mediated by heme oxygenase 1 (HO-1) [96]. Additionally, quercetin has shown the ability to suppress the atherosclerotic inflammatory processes induced by oxidized low-density lipoprotein. This involves diminishing endothelial leukocyte adhesion and reducing the expression of inflammatory mediators like IL-6, TNF-α, COX-2, and iNOS [97,98]. These actions are accomplished by mitigating the TLR and/or NF-κB signaling pathway. In a murine model of asthma induced by OVA, quercetin displayed beneficial effects by alleviating allergic airway inflammation and airway hyperresponsiveness (AHR). This was achieved through alterations in the polarization of Th1 and Th2 immune cells, attributed to its impact on the expression of T-bet 21 and GATA binding protein 3 (GATA-3) [99]. Furthermore, quercetin exhibited the potential to ameliorate lung inflammation and emphysematous changes in mice with a COPD phenotype [100].

Resveratrol is a polyphenolic stilbene compound naturally occurring in various plants such as grapes, peanuts, blueberries, and mulberries. Its potential benefits in several health aspects are evident. Moreover, resveratrol has shown promise in RA management by reducing the expression of COX-2, ROS, prostaglandin E2 (PGE2), and inflammatory cytokines and by modulating the polarization of Th17 cells [101,102]. Regarding atherosclerosis, resveratrol’s effects are noteworthy. It interferes with the TLR4-mediated inflammatory response, impedes monocyte adhesion by suppressing intercellular adhesion molecules (ICAM)-1 expression, and inhibits the differentiation of monocytes into macrophages. Resveratrol demonstrates positive effects in the context of asthma. It alleviates airway inflammation and remodeling in animal asthma models. This is linked to its regulation of the mobility group box 1 (HMGB1), TLR4, and NF-κB pathway and the modulation of Syk protein expression. In various conditions, resveratrol exhibits its potential as an activator of sirtuin 1 (SIRT). IBD benefits from resveratrol’s therapeutic effects. In both Crohn’s disease (CD) and ulcerative colitis (UC), it reduces pro-inflammatory cytokines (IL-1β, IL-6, and TNF-α) and restores the balance between Tregs and Th17 cells [103,104]. Furthermore, resveratrol demonstrates a neuroprotective role in vitro and in vivo, showcasing its potential in tackling neurodegenerative diseases.

### 4.3. Immunotherapy in Cancer

Immunomodulatory medications including checkpoint inhibitors and adoptive T-cell therapy are being researched to enhance immunological responses. Individualized approaches that consider the patient’s immune profile are necessary for successful outcomes. Aging considerably raises the risk of developing cancer, one of the most major diseases and mortalities in humans. The introduction of immune checkpoint blockade has transformed cancer treatment by producing exceptional therapeutic advantages. By removing inhibitory signals that prevent T cells from becoming activated, immune checkpoint blockade allows tumor-reactive T cells to circumvent regulatory mechanisms and launch a potent anti-tumor response [105]. However, a sizeable percentage of patients taking immune checkpoint inhibitors experience resistance, highlighting the need to investigate the mechanisms of treatment resistance and develop improved therapeutic approaches. Better patient selection and the prevention of adverse effects are required because checkpoint inhibitors have the potential to produce quickly developing illnesses and life-threatening autoimmune toxicities. Although PD-1, PD-L1, and cytotoxic T-lymphocyte–associated antigen (CTLA)-4 suppression is crucial in restoring T-cell activation, only a tiny proportion of patients benefit from checkpoint inhibitors [106]. Monoclonal antibodies that target other immunological checkpoints, like LAG3 (CD223), have gained more attention as a result of this. Although the clinical biomarkers like PD-L1 expression and tumor mutational burden, which have been used to predict outcomes, have certain limitations, there are still challenges in applying these markers to a variety of cancer types and treatment conditions [107]. To optimize individualized immunotherapeutic treatments, continuous efforts are being made to search for relatively more potent biomarkers and therapeutic modalities in addition to PD-1/PD-L1 and CTLA-4 targeting [108]. A buildup of dysfunctional T cells may occur in the body as a result of aging, persistent infections, or malignancy. Malignant tumors can cause two important dysfunctional states in T lymphocytes in cancer patients: exhaustion and senescence. Both conditions maintain a suppressive tumor microenvironment, which prevents the body’s immune system from responding effectively to the tumor [108]. According to research, human T cells experience senescence during persistent viral infections and in tumor-infiltrating lymphocytes in a variety of malignancies, including lung, ovarian, breast, and colorectal cancer [109]. This suggests that malignant tumors use T-cell senescence as a means of evading immune surveillance. Senescence has been identified as a target for immunotherapy and a peripheral predictive biomarker for immune checkpoint inhibitor therapy, in addition to other supporting findings. Important to note is that immunological response and immune evasion during immune checkpoint blockade are influenced by changes in chromatin structure, notably through epigenetic modifications. Notably, chromatin remodeling and epigenetic reprogramming continue in a sequence from functional to dysfunctional T-cell states, which ultimately result in a completely dysfunctional state [110,111]. Anti-CTLA-4 medication alters the T-cell pool implicated in tumor detection by extending the blood TCR repertoire, which causes both an increase in treatment side effects and a rise in the quantity of tumor-reactive T-cell clones. This therapy causes polyclonal proliferation of TCR clones, including those that are not specific to tumor antigens, within the tumor microenvironment. In contrast, the pre-treatment T-cell clonality inside tumors has not reliably predicted the response to CTLA-4 inhibition. Moreover, anti-PD1 therapy has been shown to increase TCR clonality, particularly in responders, while reducing TCR diversity in intra-tumoral infiltrating cells [112]. Studies have shown that higher peripheral blood TCR diversity is associated with improved outcomes for melanoma patients receiving anti-CTLA-4 or anti-PD1 therapy [113]. Changes in baseline TCR diversity have been linked to treatment outcomes in patients with metastatic urothelial cancer and gastrointestinal malignancies [114]. Although pre-treatment TCR diversity did not predict outcomes, post-treatment constraints on TCR diversity followed by increases correlated with favorable outcomes, based on recent research on patients with renal cell carcinoma undergoing anti-PD1 therapy [115]. Although there are significant differences, studies have generally linked higher blood TCR variety at baseline and enhanced TCR clonality after checkpoint blockade immunotherapy to better clinical outcomes and longer survival. These differences can be attributed to disease type, intra-tumoral mutation rate, prior treatments received by patients, and technology used to measure TCR diversity. The choice of sample used to calculate TCR diversity, particularly certain blood T-cell subsets such as peripheral PD1+ T cells, may provide relatively more information regarding the relationship between TCR diversity and the response to checkpoint blockade immunotherapy. This subgroup has demonstrated relationships between TCR diversity and therapeutic results in several cancer types and is regarded as typical of tumor-specific T lymphocytes [114,116,117].

As a result, immunological aging due to age can be managed through immune function control and cancer immunotherapy by controlling inflammation with probiotics and natural chemicals (Table 2).

## 5. Conclusions

Aging has a profound impact on the immune system, leading to a complex process known as immunosenescence. This age-related deterioration in immune function results in increased susceptibility to infectious diseases, reduced response to immunization, a higher prevalence of cancer and autoimmune disorders, and chronic inflammation associated with age-related diseases. The causes of immunosenescence include telomere shortening, altered signaling pathways such as mTOR, and the production of pro-inflammatory molecules known as the SASP. These changes contribute to the development of various age-related disorders, including cancer, atherosclerosis, and immunodeficiency disorders (Figure 1). However, age-related immunological diseases can be treated in a variety of techniques, and those approaches that are based on molecular pathways have a lot of promise. The main goals of these tactics are to reduce inflammation and strengthen the immune system to fight off numerous age-related disorders. Continued research in this field holds the potential to revolutionize the approach to immune diseases associated with aging, ultimately promoting healthier aging for individuals worldwide.

## Figures and Tables

**Figure 1 ijms-24-15584-f001:**
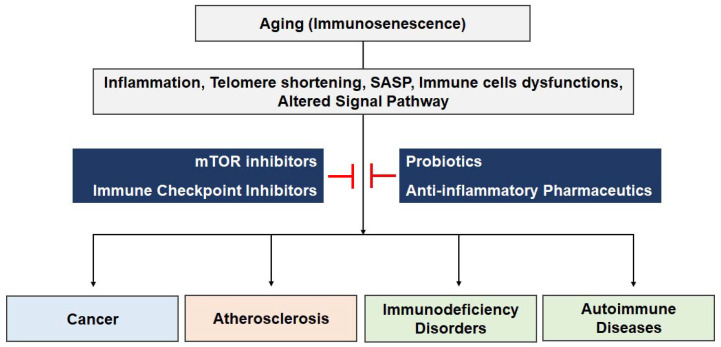
Immunosenescence and diseases. This diagram shows the several biochemical processes that are engaged during the immunological aging process carried on by the consequences of aging. The phenomenon of immune aging precipitates a spectrum of diseases, the management of which can be achieved through the manipulation of mTOR, immune checkpoint inhibitors, probiotics, and anti-inflammatory pharmaceuticals. The emergence of these modulatory agents paves the path towards the innovation of novel therapeutics functioning as immune aging modulators, aimed at the regulation of a myriad of diseases.

**Table 1 ijms-24-15584-t001:** Summary of molecular mechanisms that control immunological aging.

Molecular Mechanisms	Key Findings
Telomere shortening	- Cellular senescence is caused by telomere shortening during cell division - Telomeres are protein–DNA complexes at the ends of linear chromosomes - Various factors like ROS, inflammation, genetics, and hormones can affect telomere length - Short telomeres can lead to cell cycle arrest and senescence - Immune cells can increase telomerase to prevent telomere attrition
Immunosenescence	- Immunosenescence is characterized by changes in the immune system due to antigen exposure and oxidative stress - Aging leads to a decline in naive T and B cells and a reduction in lymphocyte numbers - The innate immune system remains relatively intact in the elderly - Neutrophils and senescent cells play roles in immune cell aging
Inflammation and SASP	- Senescent immune cells secrete pro-inflammatory substances known as the SASP, which disrupt tissue microenvironments and contribute to age-related diseases - The SASP has both positive and negative effects on the body, including inflammation and DNA damage signaling
Altered signaling pathways	- Changes in signaling pathways, especially the mTOR pathway, impact immune cell function - Dysregulated nutrient sensing and metabolism affect T-cell activation and overall immune responses - mTOR plays a significant role in regulating cellular growth and metabolism in response to nutrition and hormones - Pharmacological inhibition of mTOR can extend lifespan and influence immune cell function

**Table 2 ijms-24-15584-t002:** Summary of treatment approaches based on molecular mechanisms.

Section	Topic	Key Findings
Section 4.1	Probiotics	- Probiotics regulate inflammation by influencing immune cells, including monocytes, macrophages, DCs, NK cells, and lymphocytes - Specific probiotic strains release anti-inflammatory cytokines, such as IL-10 - Probiotics show promise in managing conditions like Crohn’s disease by affecting immune regulatory networks - Intestinal dysbiosis in aging individuals can lead to gut inflammation and affects cytokine balance
Section 4.2	Natural compounds	- Polyphenols like curcumin, quercetin, and resveratrol have anti-inflammatory properties - Curcumin inhibits NF-κB, reducing pro-inflammatory cytokines and COX-2 - Quercetin modulates cytokine balance and inhibits NLRP3 inflammasome activation - Resveratrol influences the polarization of immune cells and cytokine production - TLR4, MAPK, and NF-κB pathways are affected by these natural compounds
Section 4.3	Cancer immunotherapy	- Dysfunction in T cells hinders immune response in cancer patients - Chromatin remodeling and epigenetic modifications impact immune evasion during immune checkpoint blockade - TCR diversity in blood is related to cytokine balance and regulatory factors

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
