# Peer review of "Immune Diseases Associated with Aging: Molecular Mechanisms and Treatment Strategies"

_ijms, 2023, doi:10.3390/ijms242115584_

Round 1

Reviewer 1 Report

Comments and Suggestions for Authors

This review discusses the role of the age-dependent decline of immune functions in the development of cancer, atherosclerosis and immune deficiency, with a focus on molecular mechanisms and possible therapeutic approaches. The paper is interesting; however, some issues should be addressed before considering this manuscript for publication:

1.           The syntax is extremely complex, with a frequent use of parenthetical clauses. Some sentences (for example: page 1 lines 40-42 “Immunosenescence, an aging process that leads in a decrease in immune system function, number, and function of immunological organs, immune cells, and cytokines [3, 4].”) show some grammatical errors. The manuscript needs to be fully revised by a native English speaker.

2.           Please, kindly explain all the acronyms in the text including cytokines and chemokines.

3.           Reference should be updated: excellent recent reviews discuss the impact of aging on immune cells. Also, very recent papers analyse the use of natural compounds and new drugs in reverting immunsenescence, inflammaging and age-dependent intestinal barrier disfunction.

4.           In the Introduction or in a specific paragraph preceding the Section 2, a comprehensive description of SASP, immunosenescence and inflammaging should be added, so that established features of these senescence-associated scenarios are explained only once.

5.           Paragraph 2.1: only those aspects of immunosenescence and inflammaging influencing the onset/persistence of tumours should be discussed. General characteristics of immunosenescence and inflammaging should be discussed in the Introduction or in a specific paragraph preceding the Section 2, including all the immune cells that are analysed in detail (for example dendritic cells, that are mentioned at line 87 with no further details about how aging affects them – these effects are only partially described in paragraph 2.2). Also, a number of cytokines are cited in the paragraph 2.1. In the introduction or in a specific paragraph, the authors should briefly specify if these cytokines are pro- or anti-inflammatory and their main source in the context of SASP, immunosenescence and inflammaging.

6.           Page 2 lines 67-68 “The increase of regulatory T cells is another factor that accelerates immunological aging.”: reference is missing. Tregs are again discussed below in the same paragraph (from line 76) after mentioning a completely different subject (thymic involution). Please, kindly re-write the paragraph in order to show the information in a neater and more understandable way.

7.           Page 2 lines 75-76 “Apoptotic molecules like CD95 and costimulatory molecules like CD28 and CD27 on T cells are both decreased in the aged”. Please kindly explain why this piece of information is important in the text.

8.           Page 2 line 83 “the inflammatory environment that is characteristic of the elderly”: are the authors referring to inflammaging?

9.           Page 2 lines 85-87 “In addition, the tumor microenvironment can depress the immune system by preventing hematopoietic stem cells from differentiating into mature dendritic cells (DCs).” Is this a mechanism detected only during aging? It seems that the authors aim to describe all the immune alterations associated with tumorigenesis. If this is the case, the dissertation should be more general at the beginning of the paragraph, then becoming more specific on immunosenescence/inflammaging aspects influencing tumour onset/resistance to therapy in order to match the chosen title for the manuscript.

10.         Paragraph 2.2 lines 117-123 “The aging of naive T- and B- lymphocytes, which have a small repertoire of B- and T-cell receptors, reduces their ability to mount an effective adaptive immune response [27, 28]. Additionally, the activation of the inflammatory response is hampered by the reduction in response to TLR stimulation following ligand/or antigen binding by macrophages/monocytes and DCs. Additionally, the loss in macrophage phagocytic ability, antigen-presenting ability, cytokine generation, and chemotaxis ability hinders the clonal T-cell proliferation process, which slows the development of plaque inflammation [29].” How are these events contributing to atherosclerosis in the elderly? Please, explain in the text. Also, please avoid the reiterated use of “additionally”.

11.         Page 3 line 133 “In case of T cells, Senescent” is “Senescent” a typo?

12.         Paragraph 2.2: the authors mention some chemokines, that should be described in the light of immunosenescence and inflammaging in the Introduction or in a specific paragraph.

13.         Page 3 lines 144-147 “As reported in a recent paper the significant role of distinct Tregs subsets in influencing the development and progression of atherosclerosis. Tregs are instrumental in maintaining immune tolerance and curbing excessive immune responses. These subsets of Tregs exert their effects by mitigating the responses of effector T cells, thus impacting atherosclerosis [35].” and page 4 lines 153-155 “Additionally, the discussion delves into another category of Tregs known as inducible Tregs (iTregs). This group includes type 1 regulatory T cells (Tr1) and Tregs that produce transforming growth factor-beta (TGF-b), designated as Th3 [37]. Unlike natural Tregs, iTregs develop in peripheral tissues such as the intestine from naive or effector T cells. While they are also CD4 and CD25 positive, they do not necessitate Foxp3 for functionality. Tr1 Tregs secrete IL-10 and Th3 Tregs produce TGF-b [38]. These iTregs play a vital role in maintaining immune balance by secreting immunosuppressive cytokines and contribute to the inhibition of atherosclerotic lesion formation [39]. Both natural and iTregs hold the capacity to prevent autoimmunity and excessive immune responses by suppressing the proliferation and differentiation of naive and effector T cells. They also exert control over the functioning of various other lymphoid cell types, including B cells, macrophages, and DCs [40-42]. The passage proposes that Tregs, irrespective of their antigen specificity, can exert suppressive functions post-activation via antigen presentation”. The key features of Tregs should be presented in the Introduction or in a specific paragraph preceding the Section 2, then the aspects of Treg functions linked to atherosclerosis may be discussed in paragraph 2.2.

14.         The authors frequently use “the passage”. The meaning is not clear.

15.         Section 3: should be completely re-written, in order to focus only on age-dependent molecular mechanisms that are linked to the pathological entities (tumour, atherosclerosis and immunodeficiencies) mentioned in the Section 2. As an alternative, the authors may choose to include these pieces of information in their respective paragraphs in the Section 2, thus eliminating the Section 3. Those molecular pathways that are further deepened in the following sections (example PD-1/PD-L1) deserve a more inclusive discussion.

16.         Paragraph 4.1: the authors should explain in the Introduction or in a specific paragraph how aging alters resident intestinal flora and intestinal barrier, contributing to a poor control of systemic inflammation. Moreover, the paragraph should be divided into two separated sub-paragraphs: one dissecting experimental evidences demonstrating a role for probiotics in reverting intestinal barrier disfunction during aging, and the other focusing on natural compounds. It would be worth noting that more and more compounds have been studied in the context of immunosenescence and inflammaging, like for example polyphenols of the Mediterranean diet, but the authors do not mention none of these pieces of information.

17.         Paragraph 4.2: the title should be changed, since the authors list no strategy to counteract immunosenescence with immunotherapy, instead they focus on cancer treatment. So, the title should be changed into something like “Immunotherapy in cancer”. The scope of the paragraph is not clear. Are the authors suggesting the use of immune checkpoint inhibitors specifically in the elderly? If yes, on the basis of what experimental proofs? What are the mechanisms ruling a higher rate of checkpoint inhibition success in the elderly vs general populations? What are the “medications” involved in PD-1/PD-L1 axis and CTLA-4 blockade? Are the described data recorded only in the elderly? What about LAG-3 (mentioned only at line 463)? On the basis of the chosen title, the focus of this review should be what happens during aging; thus, the authors should carefully describe every experimental evidence in the light of what was recorded in the aged population. If the available data were found in adults only (not in the elderly) this should be clearly specified, and the authors should discuss what could be expected in aged patients on the basis of immunosenescence and inflammaging.

18.         No detail is provided about treatment of the last pathological scenario (atherosclerosis). Are there no studies focused on the treatment of atherosclerosis in the aged population?

19.         The Conclusion should be extended and provide the readers with a complete panoramic of the links bringing together all the molecular mechanisms described in the previous sections

Comments on the Quality of English Language

The manuscript should be fully revised.

Author Response

Reviewer I

This review discusses the role of the age-dependent decline of immune functions in the development of cancer, atherosclerosis and immune deficiency, with a focus on molecular mechanisms and possible therapeutic approaches. The paper is interesting; however, some issues should be addressed before considering this manuscript for publication:

Response: Thanks for your advice. We have carefully corrected and improved the quality of manuscript as many as possible.

Comment 1: The syntax is extremely complex, with a frequent use of parenthetical clauses. Some sentences (for example: page 1 lines 40-42 “Immunosenescence, an aging process that leads in a decrease in immune system function, number, and function of immunological organs, immune cells, and cytokines [3, 4].”) show some grammatical errors. The manuscript needs to be fully revised by a native English speaker.

Response: As you suggested, we made a minor correction to the parentheticals and had the entire paper professionally proofread by native English speaker (Editage editing company). Attached the editing certificate proof.

Comment 2: Please, kindly explain all the acronyms in the text including cytokines and chemokines.

Response: Thanks for your comments. In accordance to your suggestion, we have explained all the acronyms used.

Comment 3: Reference should be updated: excellent recent reviews discuss the impact of aging on immune cells. Also, very recent papers analyse the use of natural compounds and new drugs in reverting immunsenescence, inflammaging and age-dependent intestinal barrier disfunction.

Response: Thanks for suggestion. We have updated the references as requested by the reviewer.

Comment 4: In the Introduction or in a specific paragraph preceding the Section 2, a comprehensive description of SASP, immunosenescence and inflammaging should be added, so that established features of these senescence-associated scenarios are explained only once.

Response: Thanks for suggestion. We added mention of this sentence.

Comment 5: Paragraph 2.1: only those aspects of immunosenescence and inflammaging influencing the onset/persistence of tumours should be discussed. General characteristics of immunosenescence and inflammaging should be discussed in the Introduction or in a specific paragraph preceding the Section 2, including all the immune cells that are analysed in detail (for example dendritic cells, that are mentioned at line 87 with no further details about how aging affects them – these effects are only partially described in paragraph 2.2). Also, a number of cytokines are cited in the paragraph 2.1. In the introduction or in a specific paragraph, the authors should briefly specify if these cytokines are pro- or anti-inflammatory and their main source in the context of SASP, immunosenescence and inflammaging.

Response: Thanks for your comments. As requested by the reviewer, we have made this clear in the introduction and also mentioned cytokines.

Comment 6: Page 2 lines 67-68 “The increase of regulatory T cells is another factor that accelerates immunological aging.”: reference is missing. Tregs are again discussed below in the same paragraph (from line 76) after mentioning a completely different subject (thymic involution). Please, kindly re-write the paragraph in order to show the information in a neater and more understandable way.

Response: Thanks for your comments. As requested by the reviewer, rewrote the paragraph to make the information cleaner and easier to understand.

Comment 7: Page 2 lines 75-76 “Apoptotic molecules like CD95 and costimulatory molecules like CD28 and CD27 on T cells are both decreased in the aged”. Please kindly explain why this piece of information is important in the text.

Response: Thanks for your comments. We added the mention of this sentence in paragraph 2.1

Comment 8: Page 2 line 83 “the inflammatory environment that is characteristic of the elderly”: are the authors referring to inflammaging?

Response: Thanks for your comments. What we are discussing pertains to the age-related increase in inflammation, which is intricately linked to a decline in immune function.

Comment 9: Page 2 lines 85-87 “In addition, the tumor microenvironment can depress the immune system by preventing hematopoietic stem cells from differentiating into mature dendritic cells (DCs).” Is this a mechanism detected only during aging? It seems that the authors aim to describe all the immune alterations associated with tumorigenesis. If this is the case, the dissertation should be more general at the beginning of the paragraph, then becoming more specific on immunosenescence/inflammaging aspects influencing tumour onset/resistance to therapy in order to match the chosen title for the manuscript.

Response: Thanks for your comments. The immaturity of dendritic cells within the tumor environment is not exclusive to the aging process; rather, it is included due to its potential occurrence in aging. Dendritic cell immaturity is observed not only in the context of aging but also in various immunosuppressive diseases. We modified the sentences.

Comment 10: Paragraph 2.2 lines 117-123 “The aging of naive T- and B- lymphocytes, which have a small repertoire of B- and T-cell receptors, reduces their ability to mount an effective adaptive immune response [27, 28]. Additionally, the activation of the inflammatory response is hampered by the reduction in response to TLR stimulation following ligand/or antigen binding by macrophages/monocytes and DCs. Additionally, the loss in macrophage phagocytic ability, antigen-presenting ability, cytokine generation, and chemotaxis ability hinders the clonal T-cell proliferation process, which slows the development of plaque inflammation [29].” How are these events contributing to atherosclerosis in the elderly? Please, explain in the text. Also, please avoid the reiterated use of “additionally”.

Response: Thanks for your comments. We've modified the reviewer's sentence to make it more relevant and added a reference.

Comment 11: Page 3 line 133 “In case of T cells, Senescent” is “Senescent” a typo?

Response: Thanks for your comments. We fixed it.

Comment 12: Paragraph 2.2: the authors mention some chemokines, that should be described in the light of immunosenescence and inflammaging in the Introduction or in a specific paragraph.

Response: Thanks for your comments. We added the mention of this in Paragraph 2.2.

Comment 13: Page 3 lines 144-147 “As reported in a recent paper the significant role of distinct Tregs subsets in influencing the development and progression of atherosclerosis. Tregs are instrumental in maintaining immune tolerance and curbing excessive immune responses. These subsets of Tregs exert their effects by mitigating the responses of effector T cells, thus impacting atherosclerosis [35].” and page 4 lines 153-155 “Additionally, the discussion delves into another category of Tregs known as inducible Tregs (iTregs). This group includes type 1 regulatory T cells (Tr1) and Tregs that produce transforming growth factor-beta (TGF-b), designated as Th3 [37]. Unlike natural Tregs, iTregs develop in peripheral tissues such as the intestine from naive or effector T cells. While they are also CD4 and CD25 positive, they do not necessitate Foxp3 for functionality. Tr1 Tregs secrete IL-10 and Th3 Tregs produce TGF-b [38]. These iTregs play a vital role in maintaining immune balance by secreting immunosuppressive cytokines and contribute to the inhibition of atherosclerotic lesion formation [39]. Both natural and iTregs hold the capacity to prevent autoimmunity and excessive immune responses by suppressing the proliferation and differentiation of naive and effector T cells. They also exert control over the functioning of various other lymphoid cell types, including B cells, macrophages, and DCs [40-42]. The passage proposes that Tregs, irrespective of their antigen specificity, can exert suppressive functions post-activation via antigen presentation”. The key features of Tregs should be presented in the Introduction or in a specific paragraph preceding the Section 2, then the aspects of Treg functions linked to atherosclerosis may be discussed in paragraph 2.2.

Response: Thanks for your helpful comments. The sentences have been modified to fit the context, and the unnecessary introduction to Tregs has been modified.

Comment 14: The authors frequently use “the passage”. The meaning is not clear.

Response: Thanks for your comments.  As requested by the reviewer, the use of “the passage” was removed and the sentence was modified.

Comment 15: Section 3: should be completely re-written, in order to focus only on age-dependent molecular mechanisms that are linked to the pathological entities (tumour, atherosclerosis and immunodeficiencies) mentioned in the Section 2. As an alternative, the authors may choose to include these pieces of information in their respective paragraphs in the Section 2, thus eliminating the Section 3. Those molecular pathways that are further deepened in the following sections (example PD-1/PD-L1) deserve a more inclusive discussion.

Response: Thanks for your comments. We have revised the paragraph throughout as requested by the reviewer.

Comment 16: Paragraph 4.1: the authors should explain in the Introduction or in a specific paragraph how aging alters resident intestinal flora and intestinal barrier, contributing to a poor control of systemic inflammation. Moreover, the paragraph should be divided into two separated sub-paragraphs: one dissecting experimental evidences demonstrating a role for probiotics in reverting intestinal barrier disfunction during aging, and the other focusing on natural compounds. It would be worth noting that more and more compounds have been studied in the context of immunosenescence and inflammaging, like for example polyphenols of the Mediterranean diet, but the authors do not mention none of these pieces of information.

Response: Thanks for your comments. At the request of our reviewer, we wrote about aging and intestinal bacteria and inflammation, and divided them into two paragraphs according to the characteristics of the paragraph.

Comment 17: Paragraph 4.2: the title should be changed, since the authors list no strategy to counteract immunosenescence with immunotherapy, instead they focus on cancer treatment. So, the title should be changed into something like “Immunotherapy in cancer”. The scope of the paragraph is not clear. Are the authors suggesting the use of immune checkpoint inhibitors specifically in the elderly? If yes, on the basis of what experimental proofs? What are the mechanisms ruling a higher rate of checkpoint inhibition success in the elderly vs general populations? What are the “medications” involved in PD-1/PD-L1 axis and CTLA-4 blockade? Are the described data recorded only in the elderly? What about LAG-3 (mentioned only at line 463)? On the basis of the chosen title, the focus of this review should be what happens during aging; thus, the authors should carefully describe every experimental evidence in the light of what was recorded in the aged population. If the available data were found in adults only (not in the elderly) this should be clearly specified, and the authors should discuss what could be expected in aged patients on the basis of immunosenescence and inflammaging.

Response: Thanks for your comments. The reviewer is correct. We modified Paragraph 4.3 because it is not an immunotherapy that only applies to older people

Comment 18: No detail is provided about treatment of the last pathological scenario (atherosclerosis). Are there no studies focused on the treatment of atherosclerosis in the aged population?

Response: Thanks for your comments. Yes, not many studies have been reported focusing on the treatment of atherosclerosis based on immunotherapy targeting the elderly. I think it would be good to describe this part if many studies are conducted in the future

Comment 19: The Conclusion should be extended and provide the readers with a complete panoramic of the links bringing together all the molecular mechanisms described in the previous sections

Response: Thanks for your comments. The conclusion was revised in response to the reviewer request.

Reviewer 2 Report

Comments and Suggestions for Authors

  • This review article explores the connection between aging and immune diseases like tumors and atherosclerosis, discussing key molecular mechanisms. It emphasizes the importance of targeting inflammation and immunomodulation for treatment, including novel approaches and immunotherapy options to enhance immune responses in the elderly. Following are my comments:

  • The introduction is concise but lacks an emphasis on the significance of the subject matter. Numerous review articles on this topic are readily available, some of which have been cited by the authors (https://academic.oup.com/cid/article/31/2/578/299255, https://www.nature.com/articles/s41380-021-01361-1, https://onlinelibrary.wiley.com/doi/abs/10.1111/j.1432-2277.2009.00927.x). However, the introduction could benefit from a clearer delineation of the unique contribution and novelty of the current article.

  • I have reservations about categorizing tumors and Atherosclerosis as exclusively age-related diseases.

  • There are key aspects which I thought should be included, like:

    • Menopause, a significant and complex factor, can disrupt the aging process in women. It's essential to delve into the unique challenges and considerations that menopause brings to the aging experience for women.

    • While it may be somewhat outside the scope of the current review article, I came across the topic of aging related to space travel. I recommend that the authors consider adding a few lines discussing this fascinating topic, as it could pique readers' interest.

  • A lot of information from the last two sections 3 and 4 can be summarized into representative figures.

Author Response

Reviewer II

This review article explores the connection between aging and immune diseases like tumors and atherosclerosis, discussing key molecular mechanisms. It emphasizes the importance of targeting inflammation and immunomodulation for treatment, including novel approaches and immunotherapy options to enhance immune responses in the elderly. Following are my comments:

Comment 1: The introduction is concise but lacks an emphasis on the significance of the subject matter. Numerous review articles on this topic are readily available, some of which have been cited by the authors (https://academic.oup.com/cid/article/31/2/578/299255, https://www.nature.com/articles/s41380-021-01361-1, https://onlinelibrary.wiley.com/doi/abs/10.1111/j.1432-2277.2009.00927.x). However, the introduction could benefit from a clearer delineation of the unique contribution and novelty of the current article.

Response: Thank you for your helpful comment. We revised the introduction part as suggested by the reviewer

Comment 2: I have reservations about categorizing tumors and Atherosclerosis as exclusively age-related diseases.

Response: Thank you for your helpful comment. The reviewer's assessment is accurate. Tumors and arteriosclerosis are ailments that exhibit an increased incidence with advancing age. Nevertheless, these two conditions have been incorporated due to their elevated prevalence concerning age and immune response. I have consequently undertaken a comprehensive revision of the manuscript.

Comment 3: There are key aspects which I thought should be included, like:          Menopause, a significant and complex factor, can disrupt the aging process in women. It's essential to delve into the unique challenges and considerations that menopause brings to the aging experience for women.

While it may be somewhat outside the scope of the current review article, I came across the topic of aging related to space travel. I recommend that the authors consider adding a few lines discussing this fascinating topic, as it could pique readers' interest.

Response: Thank you for your helpful comment. Studies on the immunological aspects of menopausal issues and associated therapeutic methods are ongoing within the context of aging in women, but more results are still needed for comprehensive review. If such research continues to be actively conducted in the future, I will ensure to incorporate it into my next review paper.

Comment 4: • A lot of information from the last two sections 3 and 4 can be summarized into representative figures.

Response: Thank you for your helpful comment. We summarized sections 3 and 4, and added table 1 and 2.

Reviewer 3 Report

Comments and Suggestions for Authors

 The paper is comprehensive, the flow is logical and the data is presented critically. The references are relevant and recent. The cited sources are referenced correctly. Appropriate and key studies are included.

However, there are some specific comments on weaknesses of the article and what could be improved:

Major points 

The article still have some issues with the consistency. Although the paper is detailed and informative, with many mechanisms presented, the structure of the paper should be improved - more new paragraphs instead of long passages of text. It first starts with diseases and mechanisms, and then with aging mechanisms, and then therapy. The main line is missing. The aim of the study should be presented clearly at the end of the introduction, and then followed-up through the whole manuscript.

Minor points

2.1. title needs slight modification - Cancer or Tumors

Treatment strategies - this is not well formulated - aging is not "treatable"

Figure 1 - Aging is not equal to immunosenescence in the brackets. some of the supplements, mentioned in the text, are not mention as a group in the figure.

Author Response

Reviewer III

The paper is comprehensive, the flow is logical and the data is presented critically. The references are relevant and recent. The cited sources are referenced correctly. Appropriate and key studies are included.

However, there are some specific comments on weaknesses of the article and what could be improved:

Major points

Comment 1: The article still have some issues with the consistency. Although the paper is detailed and informative, with many mechanisms presented, the structure of the paper should be improved - more new paragraphs instead of long passages of text. It first starts with diseases and mechanisms, and then with aging mechanisms, and then therapy. The main line is missing. The aim of the study should be presented clearly at the end of the introduction, and then followed-up through the whole manuscript.

Minor points

Comment 2: 2.1. title needs slight modification - Cancer or Tumors

Response: Thank you for your helpful comment: We fixed it

Comment 3: Treatment strategies - this is not well formulated - aging is not "treatable"

Response: Thank you for your helpful comment: The reviewer is correct. Aging is not cured; however, we are introducing strategies to enhance disease management.

Comment 4: Figure 1 - Aging is not equal to immunosenescence in the brackets. some of the supplements, mentioned in the text, are not mention as a group in the figure.

Response: Thank you for your helpful comment: We modified the figure and also modified the overall text.

Round 2

Reviewer 1 Report

Comments and Suggestions for Authors

Thanks for addressing all the listed issues.